# The Influence of Isotropic Surface Roughness of Steel Sliders on Ice Friction Under Different Testing Conditions

**Igor Velkavrh** [1,*], **Jānis Lungevičs** [2], **Ernests Jansons** [2], **Stefan Klien** [1], **Joël Voyer** [1] and **Florian Ausserer** [1]

1   V-Research GmbH, Stadtstrasse 33, Dornbirn 6850, Austria; stefan.klien@v-research.at (S.K.);
   joel.voyer@v-research.at (J.V.); florian.ausserer@v-research.at (F.A.)
2   Department of Mechanical Engineering and Mechatronic, Riga Technical University, Kipsalas str. 6b,
   Riga 1048, Latvia; janis.lungevics@rtu.lv (J.L.); ernests.jansons_1@rtu.lv (E.J.)
*   Correspondence: igor.velkavrh@v-research.at; Tel.: +43-5572-394159-28

**Abstract:** Ice friction is affected by various system and surface-related parameters such as ice temperature, ambient air temperature and humidity, relative sliding velocity, specific surface pressures and surface texture (waviness, roughness) as well as the macroscopic geometry of the samples. The influences of these parameters cannot be easily separated from each other. Therefore, ice friction is a very complex tribological system and it is challenging to draw sound conclusions from the experiments. In this work, ice friction experiments with stainless steel samples that have different isotropic surface roughness values were carried out. Two tribological experimental setups were used: (i) an inclined ice track where the sliding velocity of the freely sliding steel samples was determined and (ii) an oscillating tribometer, where the coefficient of friction was assessed. For both experimental setups, the environmental parameters such as air temperature, relative humidity and ice surface temperature as well as the test parameters such as normal load and surface pressure were kept as constant as possible. The results of the experiments are discussed in relation to the ice friction mechanisms and the friction regimes.

**Keywords:** ice friction; friction regime; coefficient of friction; sliding velocity; surface roughness; steel

## 1. Introduction

Depending on ice and ambient temperatures, sliding velocity and surface contact pressure, different mechanisms prevail that divide ice friction into different friction regimes. An important parameter influencing friction regimes, especially in the context of the present study, is the presence and thickness of a liquid-like layer on the ice surface in relation to the roughness of the ice and the slider [1,2]. Figure 1 shows a schematic representation of the Stribeck curve for ice friction. With regard to the thickness of the liquid-like layer, three different friction regimes are typically distinguished: dry, mixed and hydrodynamic friction. Dry friction describes the sliding contact of two surfaces without any intermediate layer—friction coefficient is typically high. However, in ice tribology, such conditions are extremely rare because a thin liquid-like layer is always present on the ice surface when ice temperature is above around −35 °C (depending on the contact pressure) [3]. Mixed friction occurs when the surface temperature at some points within the contact zone rises above the melting point of ice and the thickness of the liquid-like layer is still lower than the characteristic roughness of the mating surfaces—friction coefficient decreases with the thickness of the liquid-like layer. If in the contact zone the temperature is higher than the melting point of the ice and the thickness of the

liquid-like layer is greater than the height of the mating surface asperities, the friction regime is called hydrodynamic—here, friction coefficient increases with the liquid-like layer thickness due to increased viscous friction. The liquid-like layer formation is also influenced by the contact pressure, the relative velocity of the sliding bodies and the humidity.

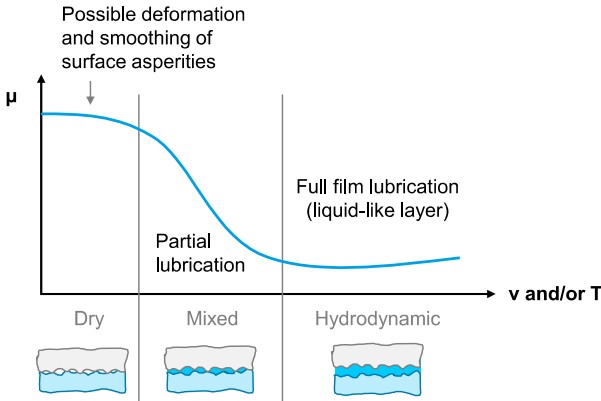

**Figure 1.** Schematic representation of the Stribeck curve for ice friction.

For different ice friction regimes, different influences of surface roughness have already been reported for polymer [4] or steel [5–9]. Under dry/mixed friction conditions, friction typically increased with increasing roughness of the slider or ice due to increased deformative friction [4–7]. On the other hand, friction under hydrodynamic friction conditions typically decreased with increasing roughness of the slider [6,8], which was ascribed to the suppression of capillary bridges. Although the asperities facilitate the formation of capillary bridges, rougher surfaces have more volume available for the propagation of meltwater compared to smooth surfaces and also result in a smaller contact area. Therefore, with decreasing roughness, the contact area and thus the adhesion as well as the friction in liquid-like layer increase. Thus, under hydrodynamic friction conditions with lower roughness, the overall friction is higher, despite the lower deformative friction component, than with higher roughness. However, for the hydrodynamic friction conditions conflicting results have also been reported. In Reference [9], smoother bobsled runners provided lower friction than the rougher ones, which was more pronounced at temperatures of −3 and −5 °C than at −10 °C. It should be noted that in the mentioned study, the coefficient of friction was calculated through the difference of initial and final sliding velocities, that is, under the presumption that higher friction results in a more pronounced reduction of velocity.

In order to adress the ambiguities present in the available literature and to gain deeper understanding of the influence of surface roughness of steel sliders under different testing conditions and/or different ice friction regimes, experimental studies were performed in the present study on stainless steel samples with different isotropic surface roughness values using two different experimental setups and test parameter sets, which has according to the knowledge of the authors previously not been performed yet. The results are discussed from the perspective of the known ice friction mechanisms in terms of friction regimes.

## 2. Materials and Methods

In this study, steel samples (Uddeholm Ramax HH) having dimensions of 35 mm × 18 mm × 14 mm (L × W × H) at a deviation of ±0.01 mm and weight of 67 ± 0.1 g were machined and their test surfaces polished with a semi-automatic polisher 334 TI 15 Mecatech (Presi, Eybens, France) to a surface roughness of Ra < 0.1 μm. Before polishing, sample sharp edges were rounded to avoid their sticking in ice during experiments. Two of the polished samples were used as reference samples. The rest of the samples additionally had their test surfaces treated by sandblasting which was followed by additional

polishing using three different re-polishing times (30, 150 and 240 s) to obtain three different isotropic surface roughness values. Additional polishing ensured that the sharp asperity peaks which formed during sandblasting were removed thereby creating relatively flat surfaces with randomly distributed valleys/dimples which act as reservoirs for the meltwater. The described surface treatment procedure was selected with the aim to verify whether this relatively affordable and well-accessible technology can be applied for effective modification of steel surfaces for sliding on ice. The surface topographies of the samples were characterized by laser scanning microscopy (VK-X250/260, Keyence International NV/SA, Mechelen, Belgium) and the macroscopic surface geometry/form of the samples were characterised by a contact type 3D profilometer (Talysurf Intra 50, Taylor Hobson, United Kingdom). Table 1 lists the surface treatments used to generate the topographies of the differently treated steel samples and their relevant roughness parameters and Figure 2 shows their surface topographies. From the Ra and Rz values (Table 1), it is clear that the applied surface treatments produced four distinct surface roughness categories. It should be noted that re-polishing after sandblasting affects the Abbott-Firestone curve and also the corresponding core roughness Rk, reduced peak height Rpk and reduced valley depth Rvk values [10]. For comparative purposes, the Rpk values describing the average height of the protruding peaks above the roughness core profile are listed in Table 1. In Figure 3, a typical macroscopic surface geometry/form of the steel samples is shown. It is clear that due to polishing, sample surfaces had a noticeable curvature which reduced the nominal contact area. Since the curvature of all samples was very similar, for the purpose of the present study, its influence on the sliding properties of the samples was not taken into account.

**Table 1.** Differently treated steel samples and their roughness values Ra, Rz and Rpk (listed in decreasing roughness order).

| Surface Treatment | Sample Number | Ra (μm) | Rz (μm) | Rpk (μm) |
|---|---|---|---|---|
| Polishing, sandblasting and additional polishing for 30 s | SP30-1 | 3.0 | 18.3 | 1.6 |
|  | SP30-2 | 2.7 | 17.5 | 1.5 |
| Polishing, sandblasting and additional polishing for 150 s | SP150-1 | 2.3 | 15.3 | 1.2 |
|  | SP150-2 | 2.0 | 14.0 | 1.1 |
| Polishing, sandblasting and additional polishing for 240 s | SP240-1 | 1.0 | 10.9 | 1.0 |
|  | SP240-2 | 0.8 | 8.4 | 0.7 |
| Polishing | P-1 | <0.1 | 1.0 | 0.2 |
|  | P-2 | <0.1 | 0.7 | 0.1 |

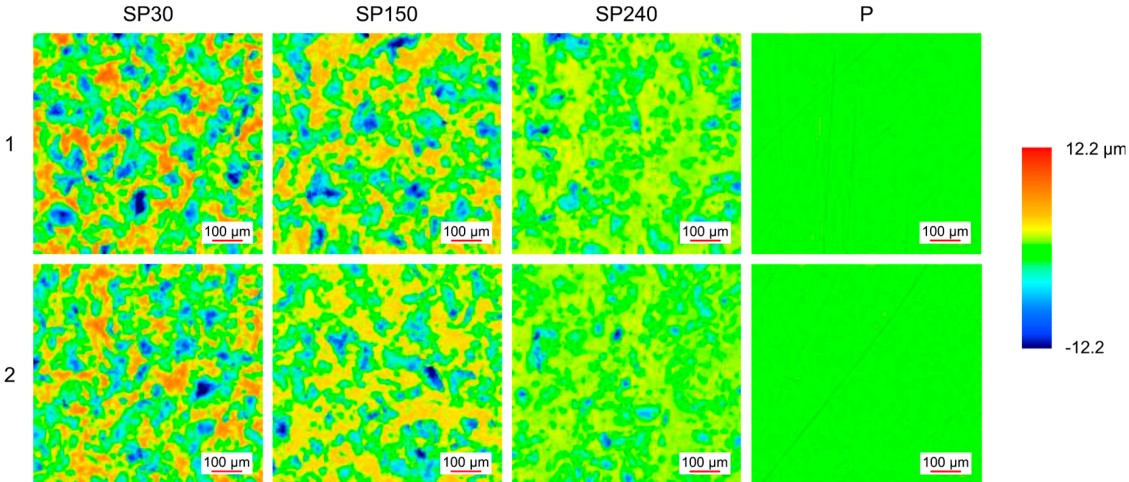

**Figure 2.** Surface topographies of differently treated steel samples.

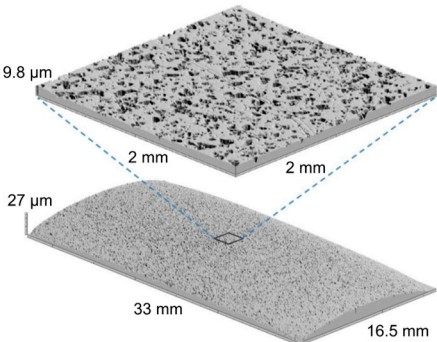

**Figure 3.** Typical macroscopic surface geometry/form of the steel samples after polishing (sample SP240-1). Magnification of a 2 mm × 2 mm area is shown for scale comparison.

Two tribological test setups were used—(i) an inclined ice track tribometer in which the steel sample slides freely after lifting the start gate while its sliding time is measured by 4 pairs of optical sensors (Figure 4a) and (ii) a modular tribometer (RVM1000, Werner Stehr Tribology GmbH, Germany) used in oscillating mode (Figure 4b).

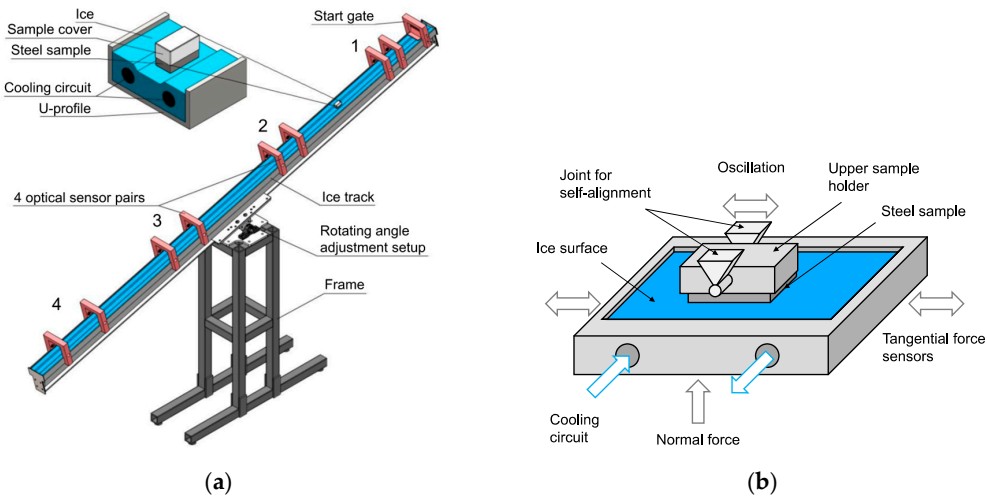

(**a**)          (**b**)

**Figure 4.** Schematic representation of the test setups used (**a**) inclined ice track tribometer, (**b**) oscillating tribometer.

*2.1. Tests on the Inclined Ice Track Tribometer*

The inclined ice track tribometer was designed as a miniature model of the skeleton field experiment [11]. It was developed to enable the comparison of the sliding velocities of steel runners from field experiments with the velocities of test samples achieved under controlled laboratory conditions. The tribometer consists of a closed, 3300 mm long U-profile with a built-in cooling system for ice formation. It is possible to achieve and maintain ice temperatures of 0 down to −20 °C which allows the simulation of different ice conditions. The tribometer is located in a climate simulation room that allows temperature control from 30 down to −20 °C. The positions of the four sensor pairs used to measure the sliding time of the steel sample were as follows (measured from the start gate, see Figure 4a): 90–140 mm, 1030–1130 mm, 1920–2070 mm and 2740–2940 mm. The steel sample thus glides over 4 different distances (with a length of 50, 100, 150 and 200 mm) over which the sliding time is measured with a precision of 1 ms. From the time measurements, four momentaneous sliding velocities were calculated. During the experiments, the ice track was tilted at an angle of 16°, which is slightly above the minimum to initiate the sliding motion of the test samples and is at the same time

sufficient to prevent from adhesive sticking of the samples in the starting position. More information on the working principle of the inclined ice track tribometer can be found in Reference [12]; however it should be noted that the latest device updates include a stronger frame, improved optical sensors and an additional climate chamber surrounding the device for minimization of the changes in temperature and humidity which occur due to the heat emitted by the researcher/operator who is present inside the climate simulation chamber during the experiments.

Before the tests, the ice surface was levelled with a specially designed planner, which can move linearly along the ice track over rolling bearings. During ice levelling, a shallow groove is formed in the middle of the ice track surface which helps to guide the sample in a straight line (see detail in Figure 4a). After levelling, remaining ice particles were carefully removed with a moist sponge and ice left untouched for a couple of minutes so it can recrystallize. Experiments were conducted at two different ice and ambient temperature conditions: (i) ice −2 °C and ambient 0 °C and (ii) ice −7 °C and ambient −4 °C. In all experiments, the relative humidity was between 60% and 80%. Temperature and humidity were controlled using 4 thermocouples (one located in air approximately 10 mm above the ice surface and the other embedded in the ice) and hygrometer (located in air approximately 10 mm above the ice). The steel samples were cooled together with the ice track before the tests in the climate simulation room. An isolating plastic sample cover was used to avoid temperature transfer from the operators' hands to the sample during the experiments. For each sample, two series of experiments were carried out in one day to minimize the influences of ambient fluctuations. Each series of experiments consisted of at least 20 measurements so that for each sample at least 40 measurements were performed at the defined ice and ambient temperatures. For the evaluation, three measurements with the highest and three measurements with the lowest velocity were not considered (Figure 5a) and the final result is calculated as the average value of the remaining measurements. In all experiments conducted on the same day, the same ice surface was used. It should be noted that during free sliding, steel sample has 3 degrees of freedom (2 linear, 1 rotational), therefore unwanted lateral translation (Figure 5b, position 2) and rotation (Figure 5b, position 3) can occur which can affect the scattering of the results. It could be observed that in the cases where the unwanted translational or rotational movement of the sample occurred (this behavior was purely stochastic), slightly lower sliding velocities were typically achieved compared to the cases where the unwanted movement did not occur. However, since in the performed experiments these deviations were smaller than the differences which occurred due to the applied surface treatments of the samples, no detailed analyses of the sample movement were conducted.

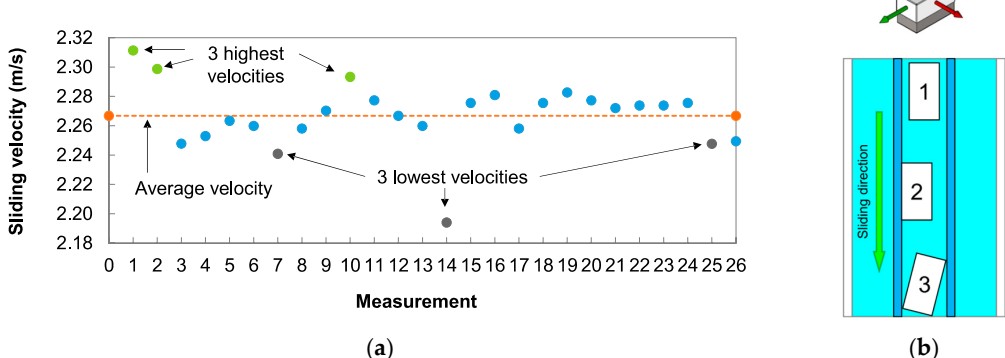

(**a**)　　　　　　　　　　　　　　　　　　　　　　　　　(**b**)

**Figure 5.** (**a**) Example of the results from a single series of experiments on the inclined ice track tribometer and their post-processing (3 highest and 3 lowest velocities were not considered); (**b**) degrees of freedom of the steel samples during tests on the inclined ice track tribometer: 1—ideal movement position, 2—unwanted lateral translation of the sample, 3—unwanted rotation of the sample.

### 2.2. Tests on the Oscillating Tribometer

The modular tribometer is equipped with an insulated test chamber, which is continuously flooded with dry, cool air and enables the establishment of the required environmental conditions (low temperatures at defined humidity). The steel sample is mounted in the upper sample holder, which allows self-alignment in the direction of movement. The steel sample moves against an ice surface having dimensions of 80 mm × 20 mm × 5 mm. To produce the ice, 18 mL of distilled water was used to which 0.5 mL of tap water was added to accelerate ice formation. Due to the expansion of the water volume at sub-zero temperatures, the ice had a convexly curved surface, so the surface was first flattened with an aluminum plate (45 mm × 28 mm) before the experiment. The smoothing was performed at a normal force of 692 N and an average sliding velocity of 0.08 m/s until the height difference between the left and right sides of the ice surface was lower than 100 μm. The flatness of the ice surface was measured with a built-in tribometer dial gauge. During the experiments, the ice temperature at the surface was −8 °C (measured with a thermocouple), with the relative humidity being 27 ± 3% and the ambient temperature being 4 ± 1 °C. These were measured with a thermometer/hygrometer inside the test chamber.

Experiments were carried out at a constant normal load of 52 N and a stroke of 24 mm. The steel samples were stored in a freezer at −18 °C for 24 h prior to testing. For each test, a run-in period of 60 s at 0.10 m/s was first performed to adjust the sample temperature to the ice temperature. Afterwards, experiments were carried out at 7 velocity levels (average sliding velocities of 0.02, 0.05, 0.10, 0.14, 0.19, 0.29 and 0.38 m/s)—during each experiment, friction measurements at all velocity levels were performed twice—once at increasing and once at decreasing velocity (Figure 6a). At each velocity level, at least 10 cycles (one cycle consisting of one forward and one backward stroke) were performed. For each sample, three experiments were carried out one after the other on the same ice surface without removing the steel sample. The first test was carried out with the test chamber closed by a transparent plastic plate so that the contact between the steel sample and the ice could be visually assessed. The second test was performed with a closed and sealed test chamber to allow even lower humidity values. The third test was carried out with an opened test chamber to allow higher humidity values. The coefficient of friction was measured continuously during the tests, while for the evaluation, 5% of the friction signal was omitted at the beginning and at the end of each stroke to eliminate the influence of static friction (Figure 6b). In Table 2, the average ambient temperatures and relative humidity values from the three sequential tests are listed and in Figure 6 modification of sliding velocities during each experiment (Figure 6a) and friction signal from a single oscillating cycle (Figure 6b) are presented.

**Table 2.** Average ambient temperatures and relative humidity values from three sequential tests conducted on the oscillating tribometer.

| Test | Ambient Temperature | Relative Humidity |
|---|---|---|
| Test 1 (chamber with plate) | 8 ± 1 °C | 41 ± 7% |
| Test 2 (closed chamber) | 4 ± 1 °C | 27 ± 3% |
| Test 3 (opened chamber) | 7 ± 1 °C | 70 ± 4% |

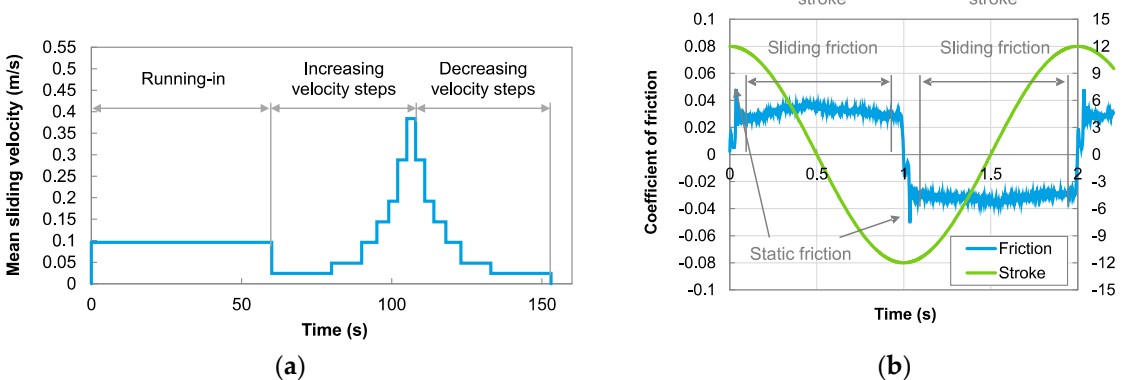

**Figure 6.** (**a**) Modification of the sliding velocities during each experiment performed on the oscillating tribometer and (**b**) measured friction signal from a single oscillating cycle.

## 3. Results

### 3.1. Sliding Velocities Measured on the Inclined Ice Track Tribometer

Figure 7 shows the velocities measured on the inclined ice track tribometer. During sliding, the sliding velocity of all samples was increasing with the sliding distance due to the accelerated movement (free sliding downwards) so that in position 1 (Figure 7a) the velocities were the lowest and in position 4 (Figure 7d) they were the highest (note that for a clearer representation of the measured differences, y-scale is different in every diagram). From Figure 7 it is clear that in all measurement positions (1-4) and at both ice and ambient temperatures, the velocity of the steel samples was approximately inversely proportional to their roughness: the samples with the highest roughness, SP30-1 and SP30-2, reached the lowest velocities, while the sample with the lowest roughness, P-1, reached the highest velocity. In the existing literature, a reduction of friction with decreasing surface roughness has typically been reported for the dry/mixed friction regimes and attributed to reduced deformative friction [4–7]. This may also be the case with the experiments conducted on the inclined ice track tribometer.

The difference between the sliding velocities of rough and smooth samples was more pronounced at low velocities (positions 1 and 2) than at high velocities (positions 3 and 4). This could be associated with higher deformative friction at lower velocities and stronger adhesion between the ice and the steel sample surfaces. Increasing velocity inhibits the formation of capillary bridges between mating surfaces thus reducing adhesion. In all measuring positions, the sliding velocities were somewhat lower at higher ice and ambient temperatures (ice −2 °C, environment 0 ± 0.5 °C) than at lower ice and ambient temperatures (ice −7 °C, environment -4 ± 0.5 °C). At the same time, the influence of roughness on velocity was somewhat less pronounced at lower ice and ambient temperatures, especially the samples with higher roughness (samples SP30 and SP150) showed a less pronounced decrease in velocity. It is worth mentioning that also in the previous research of the authors, the influence of surface texture of the sliders on their velocity typically decreased at lower ice and ambient temperatures which was true for tests on the inclined ice track tribometer [12] as well as for skeleton field tests [11]. Similar observations were made independently by the winter sports athletes who collaborated in the skeleton field tests.

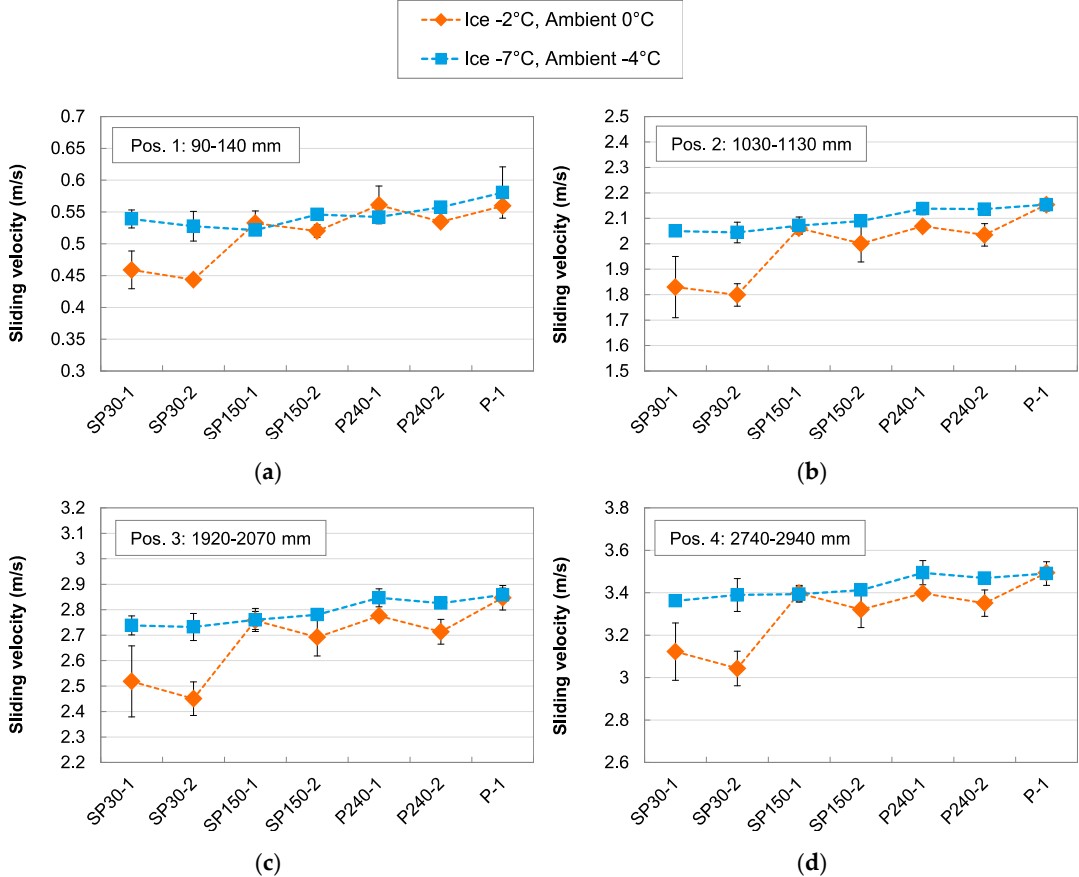

**Figure 7.** Velocities measured on the inclined ice track tribometer for 4 different measuring positions (**a**) position 1: 90–140 mm, (**b**) position 2: 1030–1130 mm, (**c**) position 3: 1920–2070 mm and (**d**) position 4: 2740–2940 mm. Results at two environmental conditions are shown: (**i**) ice temperature −2 ± 0.5 °C and ambient temperature 0 ± 0.5 °C and (**ii**) ice temperature −7 ± 0.5 °C and ambient temperature −4 ± 0.5 °C. Note that for a clearer representation of the measured differences, y-scale is different in every diagram.

### 3.2. Coefficients of Friction Measured on the Oscillating Tribometer

Figure 8 shows the coefficients of friction measured on the oscillating tribometer. It can be seen that in all tests friction increased with decreasing surface roughness. The lowest coefficients of friction of approximately 0.04 were measured with samples SP30-1 and SP30-2. With samples SP150-1, SP150-2, SP240-1 and SP240-2 slightly higher coefficients of friction of 0.06 to 0.08 were measured, while samples P-1 and P-2 provided the highest coefficients of friction of 0.1 to as high as 0.24. Assuming that lower friction leads to a higher velocity (due to lower resistance during sliding), the results from the oscillating tribometer (Figure 8) show a reverse tendency compared to the results from the inclined ice track tribometer (Figure 7). In the existing literature, an increase in friction with decreasing surface roughness has typically been reported for the hydrodynamic friction regime and attributed to higher viscous friction [6,8]. This may also be the case with the experiments conducted on the oscillating tribometer.

From Figure 8 it is also clear that for all samples, coefficient of friction was the highest in Test 1 (Figure 8a,b), slightly lower in Test 2 (Figure 8c,d) and the lowest in Test 3 (Figure 8e,f). This was the most pronounced for the smoothest samples, P-1 and P-2 and the least pronounced for the roughest samples, SP30-1 and SP30-2. Most probably, this effect is correlated with the properties of the ice surface. Since the Tests 1–3 were conducted on the same ice surface, one after another without unmounting the steel sample, the ice surface was being gradually smoothened and affected by frictional heating which enabled easier the formation of the liquid-like layer. Therefore, during Test 3, the formation of the

liquid-like layer was more efficient than during Test 1. In Test 3, where the lowest friction values were measured, most likely, the higher ambient humidity (Table 2) acted as an additional factor for efficient liquid-like layer formation.

Generally, no difference in friction was detected between the increasing and decreasing velocity levels.

In Figure 8, a slight decrease in coefficient of friction values with increasing velocity is also visible. This effect was the most pronounced for the smoothest samples, P-1 and P-2, while for the rougher samples it is almost absent (this is especially true for Tests 2 and 3). A decrease in friction with increasing sliding velocity is typically reported for the dry and mixed friction regimes, since at higher velocities more frictional heat is generated, resulting in more efficient production of the liquid-like layer which facilitates easier sliding [1,2]. In the dry/mixed friction regime the velocity dependence of the friction coefficient is described with the following equation:

$$\mu \propto 1/\sqrt{v} \tag{1}$$

where $\mu$ is the coefficient of friction and $v$ is the sliding velocity.

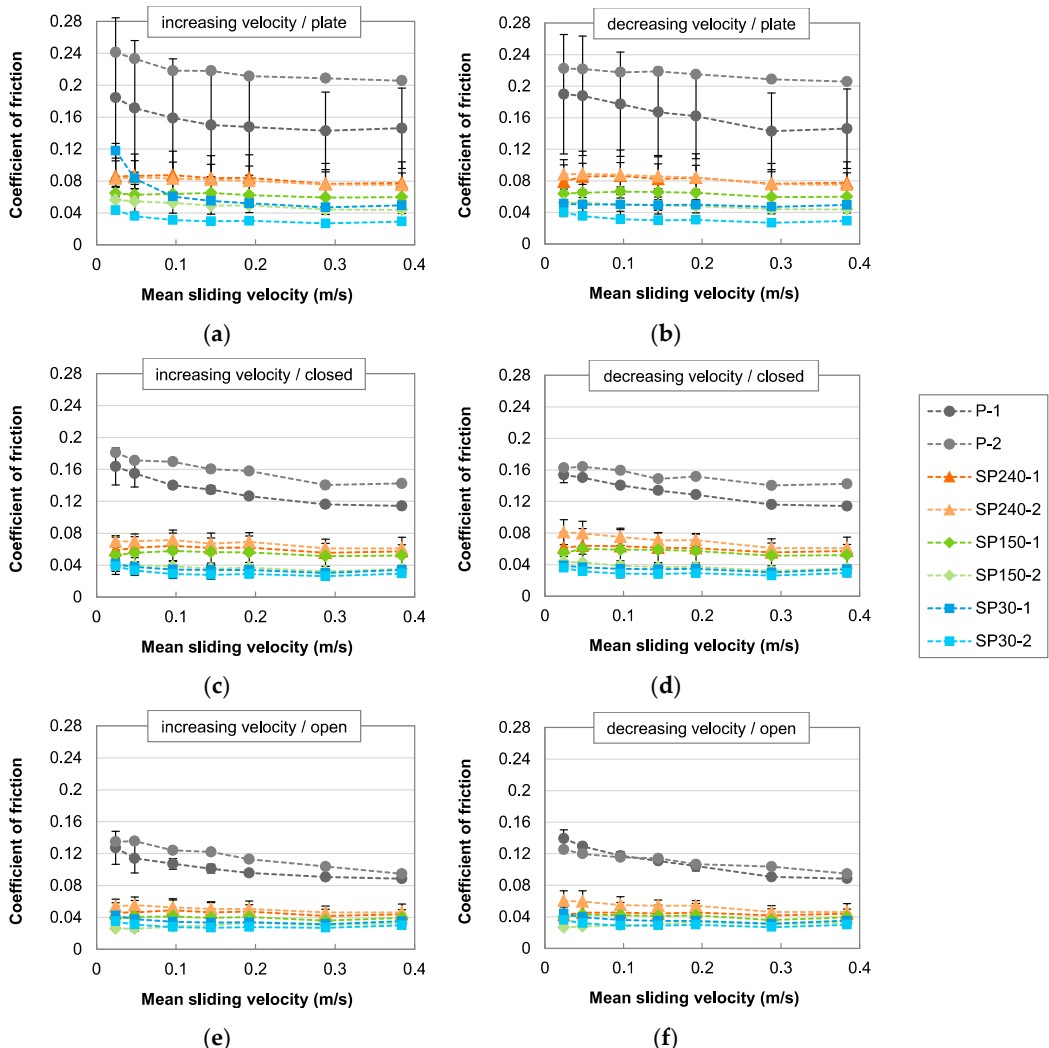

**Figure 8.** Coefficients of friction measured on the oscillating tribometer: (**a**,**b**) Test 1: test chamber closed by a transparent plastic plate, (**c**,**d**) test chamber closed and sealed, (**e**,**f**) test chamber opened; (**a**,**c**,**e**) increasing velocity levels, (**b**,**d**,**f**) decreasing velocity levels.

## 4. Discussion

Table 3 shows a comparison between the experimental parameters and the contact conditions on the inclined ice track tribometer and on the oscillating tribometer. Due to the different test parameters and sliding conditions, different friction regimes may have occurred on both test rigs. The formation of the liquid-like layer on the ice surface is largely influenced by the ambient temperature, ice temperature, surface pressure, sliding velocity and relative humidity. In the present study, the dynamics of movement may have had a strong influence on the liquid-like layer formation and the friction regime. On the inclined ice track tribometer, the steel specimen constantly in contact with a fresh ice surface and possibly the formation of the liquid-like layer (also because of the low surface pressure) was insufficient, resulting in dry or mixed friction regime. On the other hand, on the oscillating tribometer, the steel sample was in continuous oscillating contact with the ice surface, which was most likely covered with a liquid-like layer due to frictional heat, resulting in the hydrodynamic friction regime.

In Figure 9, the influence of surface roughness on the shape of the Stribeck curve for ice friction based on results from different studies (estimated on the basis of the properties of the ice surface) is presented schematically:

- In References [4–7], under dry/mixed friction conditions, friction increased with roughness. Correspondingly, in the present study on the inclined ice track tribometer (mixed friction conditions), friction increased with the roughness of the steel sliders.
- In References [6,8] under hydrodynamic friction conditions, friction decreased with roughness. Correspondingly, in the present study on the oscillating tribometer (hydrodynamic friction conditions), friction decreased with the roughness of the steel sliders.

**Table 3.** Comparison of test parameters of the inclined ice track tribometer and the oscillating tribometer.

| Test Parameter | Inclined Ice Track Tribometer | Oscillating Tribometer |
|---|---|---|
| Ice temperature | −2 and −7 °C | −8 °C |
| Ambient temperature/ relative humidity | 0 °C and −4 °C/60%–80% | 4 °C/30% |
| Sliding contact | Steel sample slides over a fresh ice surface | Steel sample slides over a run-in ice surface |
| Sliding distance | 40 tests × 3300 mm per test | 6912 mm (before that up to 47,280 mm for ice smoothening) |
| Ratio ice track length:sample length | 84 | 1.7 |
| Motion dynamics | Unidirectional, accelerating/decelerating | Oscillating, sinusoidal |
| Contact pressure | 0.001 N/mm$^2$ | 0.08 N/mm$^2$ |
| Sliding velocity | From 0 up to ~3.5 m/s | From 0 up to 0.60 m/s |

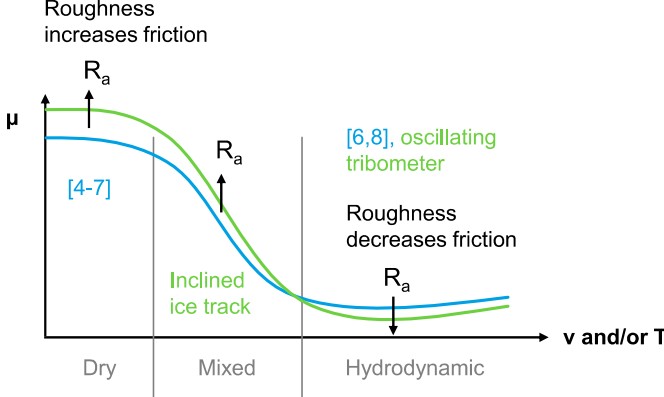

**Figure 9.** Schematic representation of the hypothetical influence of surface roughness on the shape of the Stribeck curve for ice friction based on results from different studies.

Another possible explanation for the inversely different results of coefficients of friction and sliding velocity is that under the applied testing conditions, samples with higher friction achieved higher velocities due to the more efficient formation of the liquid-like layer as a consequence of higher frictional heating. From friction between snow and waxed ski bases it is known that at low temperatures, harder waxes which prevent from snow crystal penetration (and produce more frictional heat) are needed for the enhancement of the liquid-like layer formation and the corresponding achievement of higher sliding velocities, while at higher temperatures, softer hydrophobic waxes which prevent from excessive viscous friction are used [13].

However, the hypotheses described above must still be examined in further investigations in order to be able to explain the observed effects with greater certainty.

Due to a myriad of influencing parameters, the unstable nature of ice and the still relatively vaguely understood properties of the interface that enables its low friction properties, ice friction is probably one of the most complex tribological systems with ambiguities in the understanding of the basic mechanisms. In the last decade new theoretical and experimental findings have challenged the existing theories and postulated a need for complete reformulation of the frameworks describing ice friction. As far as the authors are aware of, the present study is one of the first to compare steel sliders on ice using different experimental setups. The obtained results indicate that extreme caution is required when comparing results from different experimental setups and to some extent explains why for ice friction so many contradicting results can be found in the literature.

It should also be pointed out the nature of the liquid-like layer is much more complex than that of the Newtonian liquids considered in general lubrication theories. In recent years, novel findings contradicting the assumption that this slippery layer is a liquid are continuously being reported. The research group who made the first direct observations of this layer, prefer to call it a quasi-liquid as it represents a transitional stage between solid and liquid as the temperature increases [14]. Their observations were confirmed by complex mechanical behavior of the interstitial meltwater, which exhibits the rheology of a complex yielding material: its large viscosity, coupled to an elastic response, yields an excellent hydrodynamic lubricant behavior, leading to low friction [15]. In another study, ice slipperiness was attributed to highly mobile DA water molecules (molecules donating and accepting one hydrogen bond) that diffuse over the ice surface: in this case, a layer of mobile ice at the surface makes the surface smooth and lubricates the contact [16]. Elsewhere, authors claim that this layer should be called a "supersolid skin" because the weak bonds between $H_2O$ molecules at the surface are stretched but unlike in liquid water none of them are broken [17,18]. They also argue that this elongation of bonds ultimately produces a repulsive electrostatic force between the surface layer and anything it comes into contact with – similarly to a levitating effect.

In order to gain deeper understanding of the ice friction effects—especially in relation to the surface properties of the sliders as well as their geometry/form—further investigations are planned.

## 5. Conclusions

In the present study, analyses of sliding velocitiy and coefficient of friction of stainless steel samples having different isotropic surface roughness were conducted for ice contact on two different experimental setups and test parameter sets. It was observed that:

1. On the inclined ice track tribometer, the samples with higher roughness reached lower velocities than the samples with lower roughness, while on the oscillating tribometer samples with higher roughness provided lower friction than the samples with lower roughness.
2. Assuming that lower friction leads to a higher velocity, the results from the oscillating tribometer show a reverse tendency compared to the results from the inclined ice track tribometer.
3. In the available literature, increase of friction with increased surface roughness was typically observed for dry/mixed friction conditions due to increased deformative friction, while decrease of friction with increased surface roughness was typically observed for hydrodynamic friction conditions due to decreased adhesion and lower friction in the liquid-like layer.
4. It is possible that due to different test conditions, different friction regimes were established on both experimental setups.
5. Since the relationship between the friction and the sliding velocity is also unknown for the system under consideration, further investigations will be carried out in order to further analyze the influence of the surface roughness on ice friction in different friction regimes.
6. The presented findings indicate that due to the highly sensitive nature of ice, extreme caution is required when interpreting the results obtained under laboratory or real scale conditions in scientific research as well as in industrial practice. Conduction of comparative measurements using different experimental setups has shown to be very useful in providing a wider frame for the analysis of the ice friction mechanisms.

**Author Contributions:** Conceptualization, I.V. and J.L.; investigation, I.V., J.L. and E.J.; methodology, analysis and validation, I.V., J.L., E.J., S.K., J.V. and F.A.; writing—original draft preparation, I.V., J.L. and E.J.; writing—review and editing, S.K., J.V. and F.A.

**Funding:** Parts of this work were funded by the Austrian COMET Programme (Project XTribology, no. 849109) and carried out at the "Excellence Centre of Tribology" (AC2T research GmbH) in cooperation with V-Research GmbH and Riga Technical University. Parts of this work were also funded by the ERDF project "The quest for disclosing how surface characteristics affect slideability" (No.1.1.1.1/16/A/129) which is being implemented in Riga Technical University. Financial support of Austrian Cooperative Research (ACR) is gratefully acknowledged.

**Acknowledgments:** The authors would like to thank the Latvian bobsled and skeleton technical crew for sharing their practical observations and providing useful additional information.

**Conflicts of Interest:** The authors declare no conflict of interest.

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
