# Peer review of "The Influence of Isotropic Surface Roughness of Steel Sliders on Ice Friction Under Different Testing Conditions"

_lubricants, doi:10.3390/lubricants7120106_

Round 1

Reviewer 1 Report

The comment above the figure 1 'Deformation and smoothing of Surface asperities' might be extended by the word 'possible'.   Shown surface topographies in figure 2: the polishing after sandblasting will change the Abbott Firestone curve regarding Rpk-values. It is not necessary to show theses curves additionally but it might be helpful for the Reader to adress this fact within one sentence.  'unwanted lateral Translation of the sample', additionally described bei figure 5 (b): perfect sketch, but it would be interesting to know if the authors determined the case with highest probability or if it is pure stochastically, Suggestion: one sentence would be enough.  table 3: the aspect ratio of mating surfaces could be added to the table, this would help to understand the conclusions below.  figure 9: the word 'hypothesis' should be extended to 'schematic representation...', otherwise the reader will consider the shown influence of roughness as proven and generic.

Reviewer 2 Report

Reviewed article is very interesting and write at high scientific level. Presentation method is excellent and in accordance with generally accepted standards in that area. Figures, tables as well as terminology are clear and precise. Described method was correctly verified and compared with standard approach to this problem. Analysis are detailed and well described, scientific arguments were used to define the potential of presented method. Below are listed some comments that should be taken into consideration by the Authors to improve reviewed text:

At the end of the Introduction section authors should provide clear the aim and the novelty of the study on the basis of conclusions from state-of-the-art, not summary of presented work as it is in current form; Presented study widely covers defined scientific problem and with experimental investigations provides proper background for given conclusions, however deeper scientific consideration of obtained results referred to the basic phenomena should be given, I suggest to provide the main conclusions as numbered sentences and refer to specific values (results of analysis) as well as basic phenomena that cause described results, I suggest also to give wider description of potential use of presented findings in scientific research as well as in industrial practice.
